# Epistemic Uncertainty in State Estimation and Belief Space Planning with Learning-Based Perception Systems

**Keiko Nagami**
Department of Aeronautics and Astronautics
Stanford University, United States
knagami@stanford.edu

**Mac Schwager**
Department of Aeronautics and Astronautics
Stanford University, United States
schwager@stanford.edu

**Abstract:** Learning-based models for robot perception are known to suffer from two distinct sources of error: aleatoric and epistemic. Aleatoric uncertainty arises from inherently noisy training data and is easily quantified from residual errors during training. Conversely, epistemic uncertainty arises from a lack of training data, appearing in out-of-distribution operating regimes, and is difficult to quantify. In this work, we propose: (i) an epistemic Kalman filter (EpiKF) to incorporate epistemic uncertainty into state estimation with learned perception models, and (ii) an epistemic belief space planner (EpiBSP) that builds on the EpiKF to plan trajectories to avoid areas of high epistemic and aleatoric uncertainty. Our key insight is to train a generative model that predicts measurements from states, "inverting" the learned perception model that predicts states from measurements.

**Keywords:** Learned Perception, Belief Space Planning, Out of Distribution Data

## 1 Introduction

We consider a common architecture for vision based navigation in which a learning-based perception model ingests raw RGB images and outputs pseudo-measurements of the robot's state vector. Our research is motivated by two key shortcomings in existing methods: (i) learning covariance as an output of a perception model fails to quantify epistemic uncertainty, and (ii) learning-based perception models do not allow for projecting uncertainty into the future, posing a significant challenge to planners that reason about future uncertainty, risk, or the belief space.

To overcome these problems, we train a generative model in conjunction with the perception model. The generator takes as input the robot's state vector and outputs a predicted image of what the robot would see at that state. This is the inverse of the perception model, which takes in an image, and outputs a predicted state of where the agent took the image from.

We use the composition of the image generator and perception module in a sampling scheme to obtain an online covariance estimate for the pseudo-measurement, and apply this to the update step of a Gaussian belief filter, which we call the Epistemic Kalman filter (EpiKF). We then incorporate this filter within an optimization-based belief space planning algorithm to plan trajectories that balance efficiency with state estimation uncertainty, which we call the Epistemic Belief Space Planner (EpiBSP). The EpiBSP naturally avoids areas of high sensor noise (aleatoric uncertainty), as well as areas with sparse training data (epistemic uncertainty), without having access to the training data itself. Fig. 1 demonstrates the use of our generative model denoted $g_\phi(\cdot)$ and perception model $p_\theta(\cdot)$ to give a measurement uncertainty signal for both the current and predicted future states.

7th Conference on Robot Learning (CoRL 2023), Atlanta, USA.

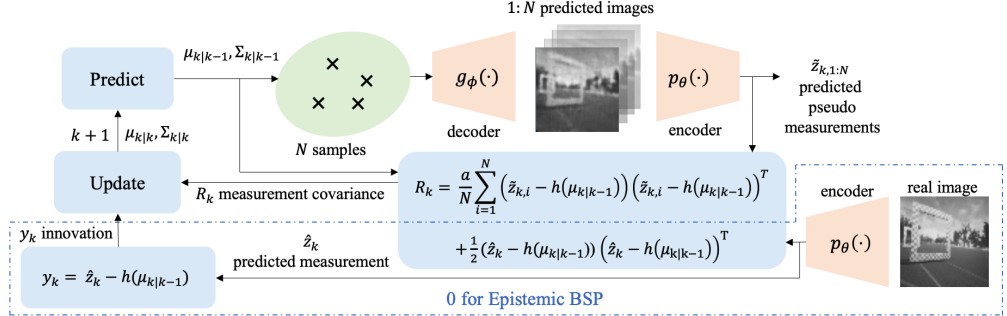

Figure 1: In both the epistemic Kalman filter and epistemic belief space planner, $N$ measurements are sampled from the predicted distribution, and passed through the neural network generator $g_\phi(\cdot)$. $N$ predicted images are forwarded to the neural network perception model $p_\theta(\cdot)$. The resulting output of predicted pseudo-measurements are used to compute the measurement covariance matrix.

## 2 Related Work

A common approach to anticipating both aleatoric and epistemic uncertainty from neural networks is using ensemble methods [1], [2], [3], [4]. However, ensemble methods require training of multiple neural networks to obtain disagreement across the models as an OOD metric. Alternative approaches for OOD detection include the methods in [5], [6], [7], and [8], however these approaches are unsuitable for our problem, since they cannot anticipate measurement uncertainty at future states as required in a belief space planner. This motivates our approach of training a measurement generator in conjunction with the perception model.

Active perception or perception-aware planning methods generally require a metric to define the quality of the perception system's performance at future states. Methods that use Visual-Inertial Odometry focus largely on aleatoric uncertainty, where the perception system's performance is tied to the amount of features in frame, as done in [9], [10], and [11], as well as their velocities or errors, as is done in [12], and [13]. Other methods in active perception have also used Neural Networks as part of the planning and control pipelines [14], [15]. However, these methods do not handle epistemic uncertainty, which is crucial in learned perception systems.

Belief space planning considers the evolution of the whole belief along future trajectories, and optimizes plans to balance trajectory efficiency with information richness. A common approach is to formulate the problem as a Partially Observable Markov Decision Process (POMDP) [16]. However, with continuous belief states, actions, and observations, the problem is typically intractable. An alternative solution is to use sampling based approaches ([17], [18], [19], [11], [20]). Another method is to use local trajectory optimization approaches ([21], [22], [23], [24]). None of these methods consider the epistemic uncertainty that arises with learned perception models.

Our work differs significantly from all of these works in the way we quantify measurement noise at future states in the plan. We use two neural networks to define our measurement uncertainty. When we compose these models, we find that the error between the input state to the generator and the output state from the perception model gives a signal that is well-correlated with epistemic uncertainty. The use of this anticipated measurement uncertainty enables us to formulate an epistemic belief space planner (EpiBSP) built on an epistemic Kalman filter (EpiKF), to account for degraded state estimation performance in the regions of high epistemic and aleatoric uncertainty.

## 3 Methods

We train a neural network perception system that takes an image input $\mathcal{I}$ and outputs a predicted pseudo-measurement of the state, $\hat{\mathbf{z}} = p_\theta(\mathcal{I})$, where $\theta$ represents network parameters. We also train an image generator to take a pseudo-measurement $\mathbf{z}$ and output a predicted image, $\hat{\mathcal{I}} = g_\phi(\mathbf{z})$,

where $\phi$ represents the network parameters. $p_\theta$ and $g_\phi$ are trained so their composition is close to the identify map on the training data, $p_\theta(g_\phi(\mathbf{z})) \approx \mathbf{z}$. We find that the error $(p_\theta(g_\phi(\mathbf{z})) - \mathbf{z})$ is a good proxy for the prediction error $(\mathbf{z} - p_\theta(\mathcal{I}))$, but has the key advantage that it does not require access to an image $\mathcal{I}$, so it can be used for planning over future states where the image is unknown. We use the pseudo-measurement to filter the state with a Kalman filter, and use this prediction error proxy to give a measurement error for the filter, to quantify OOD sources of error in the pseudo-measurement.

In the traditional extended Kalman filter equations for nonlinear systems, we have predict and update steps as follows:

$$\text{Predict:} \quad \boldsymbol{\mu}_{k+1|k} = f(\boldsymbol{\mu}_{k|k}, \mathbf{u}_k) \tag{1}$$

$$\boldsymbol{\Sigma}_{k+1|k} = \mathbf{F}_k \boldsymbol{\Sigma}_{k|k} \mathbf{F}_k^T + \mathbf{Q} \tag{2}$$

$$\text{Update:} \quad \mathbf{y}_{k+1} = \hat{\mathbf{z}}_{k+1} - h(\boldsymbol{\mu}_{k+1|k}) \tag{3}$$

$$\mathbf{S}_{k+1} = \mathbf{H}_{k+1} \boldsymbol{\Sigma}_{k+1|k} \mathbf{H}_{k+1}^T + \mathbf{R}_{k+1} \tag{4}$$

$$\mathbf{K}_{k+1} = \boldsymbol{\Sigma}_{k+1|k} \mathbf{H}_{k+1}^T \mathbf{S}_{k+1}^{-1} \tag{5}$$

$$\boldsymbol{\mu}_{k+1|k+1} = \boldsymbol{\mu}_{k+1|k} + \mathbf{K}_{k+1} \mathbf{y}_{k+1} \tag{6}$$

$$\boldsymbol{\Sigma}_{k+1|k+1} = (\mathbf{I} - \mathbf{K}_{k+1} \mathbf{H}_{k+1}) \boldsymbol{\Sigma}_{k+1|k}, \tag{7}$$

where subscript $k$ indicates the time index, $\boldsymbol{\mu}$ and $\boldsymbol{\Sigma}$ are the mean and covariance of the estimated state, and, $\mathbf{H}$ is the Jacobian of the measurement function $h(\cdot)$, $\mathbf{F}$ is the Jacobian of the dynamics equation $f(\cdot)$, $\mathbf{I}$ is an identity matrix, and $\mathbf{R}$ is the measurement noise covariance matrix. We propose to obtain $\mathbf{R}$ in (4) by first sampling pseudo-measurement vectors from the prediction distribution, passing these sampled vectors through the generator network to produce predicted images, and then passing the images through the perception network to attempt to reproduce the state vector,

$$\mathbf{z}_{k,i} \sim \mathcal{N}(\mathbf{H}_k \boldsymbol{\mu}_{k|k-1}, \mathbf{H}_k \boldsymbol{\Sigma}_{k|k-1} \mathbf{H}_k^T), \quad \forall i = 1 : N \tag{8}$$

$$\tilde{\mathbf{z}}_{k,i} = p_\theta(g_\phi(\mathbf{z}_{k,i})). \tag{9}$$

This sampling procedure is done for $N$ samples, where $i$ is the index of the sample. With the sampled measurement vectors $\tilde{\mathbf{z}}_{k,i}$, we compute their empirical covariance matrix, and combine it with the error of the estimate from the real image from the predicted mean,

$$\mathbf{R}_k = \frac{1}{2N} \sum_{i=1}^N (\tilde{\mathbf{z}}_{k,i} - \mathbf{H}_k \boldsymbol{\mu}_{k|k-1})(\tilde{\mathbf{z}}_{k,i} - \mathbf{H}_k \boldsymbol{\mu}_{k|k-1})^T + \frac{1}{2}(\hat{\mathbf{z}}_k - \mathbf{H}_k \boldsymbol{\mu}_{k|k-1})(\hat{\mathbf{z}}_k - \mathbf{H}_k \boldsymbol{\mu}_{k|k-1})^T. \tag{10}$$

This matrix $\mathbf{R}_k$ is then used as the measurement noise covariance matrix in the EKF equations. This overall pipeline is shown in Fig. 1. With this formulation we are able to get a measurement uncertainty matrix that reflects the predicted performance of the neural network perception model at the estimated state in the state space.

The trajectory optimization problem we seek to solve takes the following form,

$$\underset{\mathbf{u}_{k=1:K-1}}{\text{minimize}} \quad c_1 \text{tr}(\boldsymbol{\Sigma}_K) + c_2 \bar{\boldsymbol{\mu}}_K^T \mathbf{C}_s \bar{\boldsymbol{\mu}}_K + c_3 \mathbf{u}^T \mathbf{C}_u \mathbf{u} \tag{11}$$

$$\text{s.t. } \boldsymbol{\mu}_{k+1} = f(\boldsymbol{\mu}_k, \mathbf{u}_k), \quad \forall k = 1 : K - 1 \tag{12}$$

$$\boldsymbol{\Sigma}_{k+1} = \mathbf{F}_k \boldsymbol{\Sigma}_k \mathbf{F}_k^T + \mathbf{Q} \tag{13}$$

$$\mathbf{y}_{k+1} = \tilde{\mathbf{z}}_{k+1} - h(\boldsymbol{\mu}_{k+1}) \tag{14}$$

$$\mathbf{S}_{k+1} = \mathbf{H}_{k+1} \boldsymbol{\Sigma}_{k+1} \mathbf{H}_{k+1}^T + \mathbf{R}_{k+1} \tag{15}$$

$$\mathbf{K}_{k+1} = \boldsymbol{\Sigma}_{k+1} \mathbf{H}_{k+1}^T \mathbf{S}_{k+1}^{-1} \tag{16}$$

$$\boldsymbol{\mu}_{k+1} = \boldsymbol{\mu}_{k+1} + \mathbf{K}_{k+1} \mathbf{y}_{k+1} \tag{17}$$

$$\boldsymbol{\Sigma}_{k+1} = (\mathbf{I} - \mathbf{K}_{k+1} \mathbf{H}_{k+1}) \boldsymbol{\Sigma}_{k+1}, \tag{18}$$

where $K$ is the terminal time index, $\bar{\boldsymbol{\mu}}_K$ is the difference between the mean of the estimate $\boldsymbol{\mu}_K$ and the goal state $\mathbf{s}_g$, $\mathbf{C}_s$ is a cost matrix on the terminal state's goal, $\mathbf{C}_u$ is a cost matrix on the

Table 1: Comparison Results for Full Racetrack

|  | Gate Collisions | Error to Ground Truth | Total Time |
|---|---|---|---|
| Our Method | 0 | $0.3074 \pm 0.2315$ | 2 min 35.94 sec |
| Baseline | 1 | $0.4047 \pm 0.4453$ | 2 min 38.68 sec |

control action, $c_1$, $c_2$, and $c_3$ are constants, $\tilde{\mathbf{z}}$ is the predicted measurement, and $\mathbf{R}$ is the predicted measurement uncertainty. For the predicted measurement uncertainty, we follow a similar procedure to the measurement uncertainty in the EpiKF. However, the key challenge in the belief space planner is the lack of access to the collected image measurements from the robot. As such, we use only the first term in Equation 10 to define the measurement uncertainty in the belief space planner:

$$\mathbf{R}_k = \frac{1}{N} \sum_{i=1}^{N} (\tilde{\mathbf{z}}_{k,i} - \mathbf{H}_k \boldsymbol{\mu}_{k|k-1})(\tilde{\mathbf{z}}_{k,i} - \mathbf{H}_k \boldsymbol{\mu}_{k|k-1})^T. \tag{19}$$

Since there is no real image collected for the planner, we use a Maximum Likelihood Observation. With this formulation, our belief state dynamics are a function of the neural network generator and perception model. For each sample, we use Pytorch [25] to compute gradients of the cost with respect to the decision variables to obtain an optimized trajectory.

## 4 Track Simulation

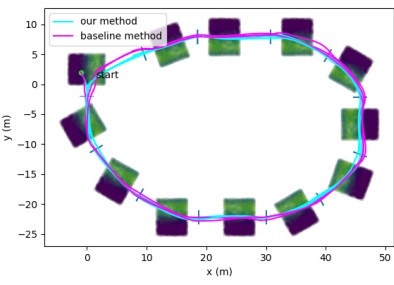

Figure 2: Trajectories of our method and baseline method through the Airsim racetrack for three laps, with light-dark domain ahead of each gate.

To showcase both the EpiKF and EpiBSP working together, we apply both to a case where a drone is flying through a racetrack with multiple gates. We use the generator and perception neural networks trained with epistemic uncertainty, where data from one side of the gate is withheld. For planning, we apply the EpiBSP by solving for a trajectory at each gate transition. Once the trajectory is optimized, we track this trajectory in a model-predictive control (MPC) loop. The resulting trajectories and light-dark domain are shown in Fig. 2. We additionally compare our method to a baseline similar to that of [26], where we use a neural network that estimates a pseudo-measurement and an associated covariance for estimation. For baseline planning, we use a trajectory from an IPOPT solver. The trajectory for three laps is also shown in Fig. 2.

To quantify the performance of our method, we compare the number of gate collisions, error of the state estimate mean to the ground truth, and the total number of time steps taken to reach the goal. These metrics are shown for our method and the baseline in Table 1. We find that our method is able to complete three laps with fewer collisions, and maintain a lower error from the ground truth than the baseline approach.

## 5 Conclusion

In this paper we demonstrate a novel approach to quantifying neural network uncertainty, and use this metric to anticipate regions of high uncertainty with belief space planning and Kalman filtering. We found that our method is able to avoid regions of both aleatoric and epistemic uncertainty to produce trajectories where the perception model is more likely to succeed. In future work, real-time belief space planning trajectories can be prioritized by applying a sampling based planning approach to reduce computation time required for the presented local optimization approach, which would enable real-world experiments.

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
