# OpenReview forum: "Epistemic Uncertainty in State Estimation and Belief Space Planning with Learning-Based Perception Systems"
_robot-learning.org/CoRL/2023/Workshop/OOD — OOD Workshop @ CoRL 2023_

### Official Review · Reviewer_uZRo · 2023-10-16
**Interesting idea to incorporate epistemic uncertainty in state estimation, but questions on calibration of uncertainty.**

**Rating:** 7
**Confidence:** 3

**Review:**

Summary:

The paper proposes a technique to account for both aleatoric and epistemic uncertainty in the setting of learned visuomotor planning. Specifically, rather than learn only a forward perception model mapping input images to measurements together with an estimate of aleatoric uncertainty in these measurements, the authors propose to learn a forward mapping as well as an inverse mapping (generator) mapping measurements to images. In this way, the reconstruction error of going from measurement space to image and back to measurement space serves as a proxy to quantify both aleatoric and epistemic uncertainty. The authors propose using the variance of the reconstructed samples as the measurement noise model in an EKF, and also propose planning in the EKF belief space to account for both aleatoric and epistemic uncertainty in their planning.

Review:

Strengths:

- The idea is interesting and a novel use of reconstruction-error based quantification of epistemic uncertainty in an EKF state-estimation framework.
- The paper clearly explained the topic, and the figure was clear and easy to understand.
- Accounting for epistemic uncertainty when planning is an important problem worthy of more research attention, and this paper proposing a promising step in that direction.

Weaknesses:

- Some parts of the description were hard to follow; e.g. it would be useful to specify what precisely was the state $x$ and measurement function $h(\mu)$ was used in the experiments?
- Equation (8) and (10) have the Jacobian $H$ multiplying the mean $\mu$ -- this strikes me as only correct if the measurement function is linear. Should this be $h(\mu)$? Including a derivation of these equations would strengthen the paper.
- What motivates combining the reconstruction residuals of the pseudo-observations along with the error wrt to the predicted mean when computing $R$ in equation (10)? I believe that when computed for $R_{k+1}$, that term amounts to $y_{k+1}y_{k+1}^T$. Given the lack of theoretical motivation, an empirical evaluation against an ablation without this additional term would be helpful to justify its inclusion.
- An additional study demonstrating the calibration of the empirical noise covariance estimated by equation (10) and the true measurement noise (both on in- and out-of-distribution inputs) would greatly strengthen the paper.

---

### Official Review · Reviewer_uaqM · 2023-10-16
**Nice way of incorporating uncertainty estimation in a POMDP formulation for perception model**

**Rating:** 8
**Confidence:** 4

**Review:**

The idea of using an inverse perception model to help examine the uncertainty of a perception model that is estimating a state from an observation is very interesting. The paper has a principled formulation of the POMDP problem by leveraging algorithmic priors such as the Kalman filter and trajectory optimization. This allows for prior assumptions on the problem such as knowledge of dynamics and the environment to be naturally built in. Additionally, the results seem very promising.

Some questions on this work relate to the quality of the uncertainty estimation proposed. How are the p and g models trained; for example, are they coming from an autoencoder model or a GAN? One can imagine that the quality and stability of the training will affect the quality of the reconstruction p(g(z)) - z. How well-calibrated is this as a proxy for the prediction error z - p(I)? Additionally, since p(g(z)) - z is the quantity that we care about, could that be incorporated into the algorithm? Currently, the covariance of the p(g(z))’s is used to capture uncertainty but not the mean of p(g(z)) - z itself. It would also be interesting to examine how the proposed covariance matrix captures epistemic and aleatoric uncertainty individually – perhaps one is expressed more than the other?

On line 92, it is stated that “the key challenge in the belief space planner is the lack of access to collected image measurements from the robot … since there is no real image collected for the planner, we use a Maximum Likelihood Observation”. What is meant by this? Why not use the generator to “imagine” future observations and use those during planning? There are prior works in the POMDP literature that do this.

For the experiment, the authors are encouraged to include in their figure a visualization of where on the Airsim racetrack the regions with low associated training data are. What is z for the experiment? As a next step, it would be very compelling to see experiments with real-world image data with strong examples of both aleatoric and epistemic uncertainty.

Minor comment: on line 68, the expression (z - g_theta(I)) seems that it should be (z - p_theta(I)).

Overall, this is a nice work using POMDP methodologies to consider the uncertainty from a learning-based module used in a control loop.

---

### Decision · Program_Chairs · 2023-10-17

**Decision:**

Accept

**Comment:**

We agree with the reviewers’ assessment that this work is technically sound and will contribute to productive, topical discussions at the 2023 Workshop on OOD Generalization in Robotics. In particular, we appreciate that the problem of epistemic uncertainty quantification is directly relevant to understanding/improving performance when operating OOD. We recommend the authors incorporate the reviewers’ feedback into their camera-ready submission to further improve their manuscript.